# Gaussian-Prior Pinwheel Convolution and Region-Energy Loss for Robust Infrared Small Target Detection

## Abstract

In recent years, convolutional neural network (CNN)-based approaches have achieved notable progress in infrared small target detection. However, most existing methods rely on standard convolution operations, which fail to capture the unique spatial distribution characteristics of infrared small targets. To overcome this limitation, we propose Gaussian-Prior Pinwheel Convolution (GPConv), a novel module that replaces standard convolutions in the lower layers of the backbone to better model the Gaussian-like spatial distribution of dim targets while enlarging the receptive field with only marginal parameter overhead. Furthermore, conventional loss functions that combine scale and localization terms often overlook the varying sensitivity across different target sizes. To address this issue, we design a Region Energy-Based Loss that incorporates a dynamic small object-aware weighting factor $\gamma(A)$ and a center distance penalty to enhance robustness across scales. In addition, we introduce a neuron-level 3D attention mechanism that jointly considers channel, spatial, and depth dimensions to refine feature representations more effectively than channel-only or spatial-only modules. Extensive experiments on the IRSTD-1K and SIRST-UAVB datasets demonstrate that integrating GPConv, Region Energy-Based Loss, and 3D attention into modern detection frameworks (YOLOv8n and RetinaNet) yields consistent and significant improvements, validating the effectiveness and generalization of the proposed approach.

## 1 Introduction

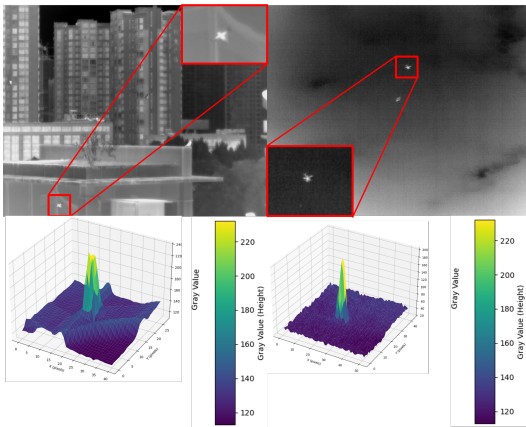

Figure 1: Grayscale 3D view of IRST

Infrared small target detection (IRSTD) has garnered significant attention in recent years due to its critical role in military and civilian applications Zhao et al. (2022b). Leveraging characteristics such as thermal sensitivity, passive radiation, strong anti-jamming performance, and excellent operability

under low-illumination conditions, IRSTD systems are widely deployed in scenarios including early warning systems for aircraft and birds, missile guidance, and maritime rescue operationsEysa & Hamdulla (2019). These tasks often require intermediate to extended range surveillance, resulting in small, low-contrast targets with limited texture and detail because of the attenuation of infrared radiation over distance Tong et al. (2024). Consequently, infrared targets typically exhibit a low signal-to-noise ratio (SNR) and a low signal-to-clutter ratio (SCR), which complicates their detection and segmentation Wang et al. (2023a). Moreover, infrared targets can vary in size, shape, and appearance due to changes in distance, motion, and observation angles Wang et al. (2023b). Compounding these challenges, complex backgrounds, such as urban structures, clouds, sea clutter, and vegetation, often produce high-intensity clutter, further masking target signatures and increasing the difficulty of accurate detection Zhang et al. (2025). Therefore, developing robust, adaptive, and real-time IRSTD algorithms remains a pressing research focus.

IRSTD techniques are generally categorized into two paradigms: traditional model-driven approaches and data-driven deep learning (DL)-based methods. Traditional methods often rely on manually crafted priors and hand-tuned parameters, such as local contrast, filtering, and background subtraction strategies, making them highly sensitive to noise and background variations Eysa & Hamdulla (2019). These approaches, while computationally efficient, exhibit limited adaptability to diverse infrared scenes and tend to suffer from low robustness in complex environments. Conversely, DL-based methods harness large-scale infrared datasets and optimize model parameters through gradient-based learning, significantly enhancing generalization and performance. Recent efforts have predominantly utilized convolutional neural networks (CNNs) to tackle IRSTDS tasks, which can be further subdivided into detection-based frameworks Wu et al. (2024). While many studies pursue performance improvements through complex architectural innovations, our approach revisits and refines the foundational convolutional module to enhance IRSTD accuracy and robustness under practical constraints.

As shown in Fig. 1, the 3D grayscale distribution of infrared small targets (IRST) reveals a Gaussian-like shape. Based on this observation, we propose a plug-and-play Gaussian-Prior Pinwheel Convolution (GPConv) module that aligns more closely with IRST imaging characteristics. Compared to standard convolution, GPConv enhances low-level feature extraction and effectively enlarges the receptive field, improving detection of small targets.

Fig. 2 illustrates that, due to the dim and small nature of IRST targets and the subjectivity involved in manual labeling, both bounding box (BBox) and mask annotations suffer from considerable IoU fluctuation errors. Although methods such as distance IoU (DIoU)Zheng et al. (2020) and complete IoU (CIoU)Du et al. (2021) losses for BBox labels, enhance IoU loss by incorporating positional information, they still fail to address the issue of IoU instability and the varying sensitivity to scale and location across targets of different sizes. To address this, the IR-SOIoU loss enhances traditional IoU by incorporating a small object-aware weighting mechanism $\gamma(A)$ and a center distance penalty term, which amplifies the impact of IoU errors for small objects—addressing IoU's insensitivity to small boxes—and helps maintain better alignment between predicted and ground truth boxes beyond mere overlap.

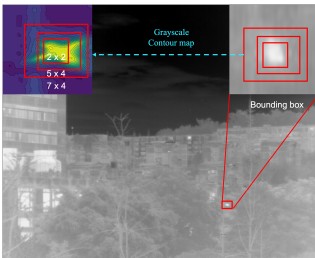

Figure 2: Visualization of Localization Errors in Bounding Box Detection

The core contributions of this paper are as follows:

- We propose GPConv, a novel plug-and-play convolutional module designed to enhance CNNs' ability to extract and analyze bottom-layer features, based on the Gaussian spatial distribution characteristics of IRST targets.

- We introduce a region energy-based dynamic loss that incorporates an area-sensitive term $\gamma(A)$, which amplifies the impact of IoU errors for small objects. This design enhances the network's regression accuracy and improves detection performance across targets of varying scales.

- We integrate GPConv and IR-SOIoU Loss into both bounding box formats within IRSTDS frameworks, validating their effectiveness and generalization on public datasets as well as our own. Experimental results show significant and consistent improvements in detection performance.

## 2 RELATED WORK

IRST detection plays a critical role in applications such as remote sensing, surveillance, and aerospace tracking. These tasks are challenging due to the low contrast, small scale, and complex background noise that often obscure targets. To address this, recent work has advanced various deep learning-based detection networks.

Traditional methods relied on handcrafted features and filtering techniques, but their limitations in complex backgrounds led to the evolution of neural-based models. One representative early work, RISTDnet, enhances robustness by using multi-scale feature fusion and context refinement strategies Hou et al. (2021). The ISNet architecture introduces a shape-sensitive detection framework, leveraging Taylor finite difference-inspired edge blocks and dual-orientation attention mechanisms to emphasize the geometric structure of small targets Zhang et al. (2022). Another direction is attention mechanisms. The Dense Nested Attention Network (DNANet) refines spatial context by embedding multi-level nested attention modules, effectively improving detection in cluttered scenes Li et al. (2022). Similarly, the Interior Attention-Aware Network (IAANet) applies a coarse-to-fine detection strategy by integrating a region proposal network with fine-grained attention modules Wang et al. (2022). More recently, ISTDet proposes an end-to-end efficient framework that compresses the detection pipeline while maintaining accuracy, and ALCNet focuses on enhancing local contrast with contextual awareness Ju et al. (2021). For a broader perspective, Cheng et al. (2024)Cheng et al. (2024) provide a comprehensive review that classifies detection networks based on key challenges such as representation, enhancement, and attention.

Loss functions play a pivotal role in the performance of IRSTD, as they directly influence the training dynamics and final detection accuracy. Unlike generic object detection, IRSTD faces challenges like extremely low signal-to-noise ratios, scale variability, and target sparsity, requiring specialized loss design. One notable advancement is the Scale and Location Sensitivity Loss, proposed by Liu et al. (CVPR 2024), which enhances detection robustness by making the loss function responsive to both scale and spatial distributions of small targets Liu et al. (2024). Pinwheel-shaped Convolution with Scale-based Dynamic Loss (SD Loss) introduces a novel strategy to mitigate intersection-over-union (IoU) fluctuations, a known issue in detecting sparsely distributed tiny targets Yang et al. (2025). A comprehensive comparison by Chen et al. (2022) evaluated BCE, IoU, and soft-IoU losses, concluding that hybrid formulations offer better generalization for various IR scenarios Chen et al. (2022). Moreover, several works propose weighted loss functions or feature-specific penalties (e.g., ResTNet's thermal-weighting loss) to prioritize salient thermal cues in complex backgrounds Zhao et al. (2022a).

IRSTD is closely tied to the quality and diversity of available datasets. Due to the nature of IR targets—often being sparse, small, and embedded in complex backgrounds—specially curated datasets are essential for benchmarking detection algorithms. One of the most widely used datasets is NUDT-SIRST, designed to evaluate infrared small target detection under various cluttered backgrounds, target morphologies, and illumination settings. It supports comprehensive testing across scenarios Li et al. (2022). The IRSTD-1K dataset was introduced with the ISNet framework, featuring diverse target scales and shapes, and has been used in several works to validate algorithm generalizability Zhang et al. (2022). To address the lack of high-density motion scenarios, DISTG was proposed as a synthetic generation algorithm producing dense infrared target sequences. It aims to facilitate training for dense target detection models and provides a new benchmark for evaluating performance in crowded scenes (Chen et al., 2024)Chen et al. (2024). Another real-world dataset, NCHU-Seg, contains 590 manually labeled infrared images. This dataset is distinguished by its inclusion of noise prediction and multi-source information fusion benchmarks, aiding in evaluating robustness Meng

et al. (2023). Additionally, Dai et al. (WACV 2021) proposed a public benchmark with asymmetric contextual modulation, focusing on real-world targets with high-quality annotations and diverse environmental settings Dai et al. (2021).

# 3 METHODOLOGY

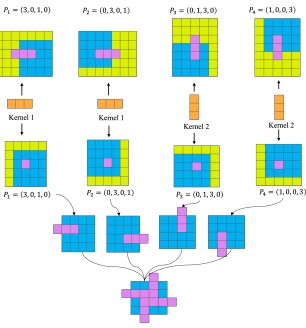

Figure 3: Pinwheel-shaped receptive field

## 3.1 GAUSSIAN-PRIOR PINWHEEL CONVOLUTION WITH ATTENTION

The pinwheel-shaped receptive field exhibits a Gaussian distribution, with its effectiveness diminishing outward. This design enhances feature extraction for infrared small targets by aligning with their spatial distribution and expanding the receptive field with minimal additional parameters Yang et al. (2025). As shown in Fig. 3, Pinwheel-shaped convolution employs asymmetric padding to generate distinct horizontal and vertical convolutional kernels tailored to different regions of the image. These kernels diffuse outward. To improve training stability and accelerate convergence, batch normalization (BN) and the sigmoid linear unit (SiLU) activation function are applied following each convolution operation. In the first layer of , parallel convolutions are performed as follows:

$$I_i\left(h, w, c\right) = SiLU\left(BN\left(I_{P_i}^{h',w',c'} \otimes K_i^{(1,3,c)}\right)\right) \tag{1}$$

where $\otimes$ denotes the convolution operator, and $W_i^{(1\times3\times c)}$ represents a $1\times3$ convolution kernel with c output channels. The padding parameters P(1, 0, 0, 3) specify the number of pixels padded to the left, right, top, and bottom of the input feature map, respectively.

Therefore, the architecture of the GPConv module is designed based on the pinwheel-shaped receptive field and shown in Fig. 4. Unlike standard convolution, GPConv employs asymmetric masks to generate pinwheel-shaped convolution kernels that focus on different regions of the image.

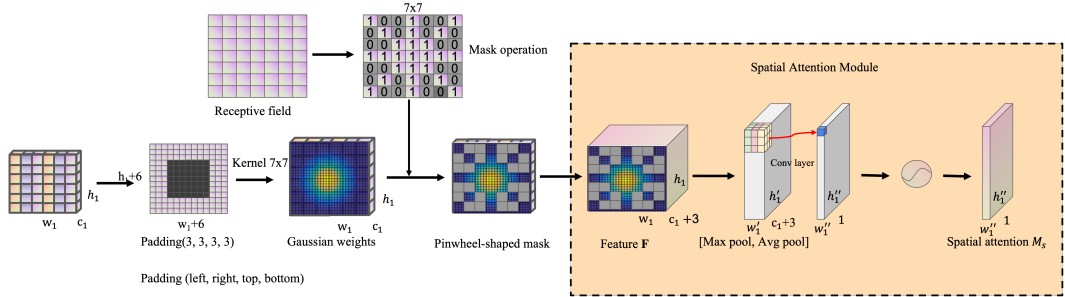

Figure 4: Gaussian-Prior pinwheel convolution with spatial attention. Best viewed in color.

Based on the Gaussian distribution characteristics of the gray levels in infrared small targets, a Gaussian kernel is employed to perform weighted averaging on the surrounding pixels, thereby

enhancing the local gray-level contrast. The 2D Gaussian function for Gaussian kernel is computed as:

$$G(x,y) = exp\left(-\frac{x^2 + y^2}{2\sigma^2}\right) \tag{2}$$

where Gaussian kernel is evaluated on a symmetric grid $[-k//2, k//2]$. The standard deviation $\sigma$ is automatically estimated using a classical empirical formulaPodobnik et al. (2008):

$$\sigma = 0.3\left(\frac{k-1}{2} - 1\right) + 0.8 \tag{3}$$

This ensures the Gaussian kernel has negligible values near the boundaries and smoothly expands with increasing kernel size $k$.

A pinwheel-shaped binary mask is constructed by setting the elements along the main diagonal, anti-diagonal, and the central horizontal and vertical lines to 1, while all other elements are set to 0.The mask matrix $M \in \{0,1\}^{k \times k}$ is defined as:

$$M_{ij} = \begin{cases} 1, & \text{if } i = j, & \text{(main diagonal)} \\ 1, & \text{if } i + j = k + 1, & \text{(anti-diagonal)} \\ 1, & \text{if } i = \frac{k+1}{2}, & \text{( horizontal line)} \\ 1, & \text{if } j = \frac{k+1}{2}, & \text{( vertical line)} \\ 0, & \text{otherwise}, \end{cases} \tag{4}$$

where $i, j = 1, 2, \ldots, k$, with indices starting from 1. Apply the mask to the Gaussian kernel:

$$W_{i,j} = G(i,j) \cdot M_{i,j} \tag{5}$$

Resulting in a kernel matrix W that retains weights only along the pinwheel directions. To ensure that the convolution kernel has an approximate DC gain (zero-frequency response) of 1, the weight matrix must be normalized.This normalization prevents any overall energy shift after convolution.

$$\hat{W} = \frac{W}{\sum_{i=1}^{k}\sum_{j=1}^{k} W_{i,j}} \tag{6}$$

Performing 2D spatial max pooling and avg pooling on a grayscale channel, the $s$ refers to the step size by which the pooling window moves across the spatial dimensions (height and width) each time.

$$3h_1 = h_1^{'}//s = h_1^{''} \tag{7}$$

$$3w_1 = w_1^{'}//s = w_1^{''} \tag{8}$$

The upper-right portion of Fig. 3 illustrates that the receptive field of PConv (with k = 3) is 25. The number of convolution operations decreases progressively from the center outward, forming a pattern similar to a Gaussian distribution. Notably, GPConv employs grouped convolution (Zhang et al., 2017), which significantly enlarges the receptive field while keeping the number of parameters minimal. The number of parameters for the convolution operation is calculated as follows:

$$Conv_{params} = \frac{C^2}{g} k^2 \tag{9}$$

where C is the number of input/output channels (assuming $C_1 = C_2 = C$), $k$ is the kernel size, and $g$ is the number of groups. This term is included only when trainable GPConv is enabled. And our GPConv's parameters are calculated as follows:

$$GPConv_{params} = \left[\frac{C^2}{g} k^2\right]_{(1)} + 2C + 98 \tag{10}$$

The term $2C$ refers to the parameters introduced by the Batch Normalization layer, comprising both the scaling ($\gamma$) and shifting ($\beta$) parameters for each of the $C$ channels. The constant 98 corresponds to the number of parameters in the spatial attention module, which utilizes a convolutional layer with a kernel size of $7 \times 7$. This layer takes two input channels—obtained via channel-wise max

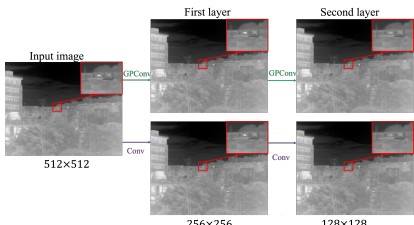

Figure 5: Visual Comparison of Feature Maps Generated by GPConv and Conv

pooling and average pooling—and produces one output channel, resulting in $2 \times 1 \times 7 \times 7 = 98$ learnable parameters.

Furthermore, the mean values across multiple channels from the outputs of both GPConv and conventional convolution (Conv) were calculated to produce the visual representations presented in Fig. 5. These visual results substantiate the effectiveness of PConv in enhancing the contrast between IRST targets and the background, while concurrently suppressing background clutter and noise-like artifacts.

## 3.2 REGION ENERGY-BASED DYNAMIC LOSS

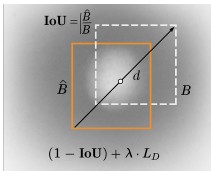

Figure 6: Pinwheel-shaped receptive field

As shown in Fig. 6, the Infra-Red Small-Object Loss (IR-SOIoU) exhibits fluctuations of up to 86%. This instability is more severe for smaller targets, which negatively impacts model stability and degrades regression performance.However, traditional IoU-based losses struggle with small objects—minor localization errors can cause IoU to drop near zero, leading to vanishing gradients. Given a predicted bounding box $\hat{B} = \left(\hat{x}, \hat{y}, \hat{w}, \hat{h}\right)$ and a ground truth bounding box $B = (x, y, w, h)$, the Intersection over Union (IoU) is defined as:

$$\text{IoU} = \frac{|\hat{B} \cap B|}{|\hat{B} \cup B|} \tag{11}$$

let d represent the Euclidean distance between the centers of the predicted and ground truth boxes, and $c$ denote the diagonal length of the smallest enclosing box that contains both $\hat{B}$ and $B$. Additionally, we define the region coverage ratio $A$ as:

$$A = \frac{wh}{A_{\text{img}}} \tag{12}$$

where $A_{\text{img}}$ is the total area of the image. This ratio reflects the relative size of the ground truth object in the image, which is particularly important for small-object detection tasks.

To address this, an area-adaptive exponent $\gamma \propto 1/\sqrt{\text{area}}$ is applied to IoU, enhancing gradients when IoU is low. Additionally, a center-distance loss $D_{\text{loss}}$ weighted by object size is introduced to explicitly penalize localization errors. For infrared images with high-contrast "hotspots," an optional region energy weight $W_e$ can further emphasize high-SNR targets.The Area adaptive index $\gamma$ are calculated as follows:

$$\gamma = \left(\tfrac{A_{\text{ref}}}{A+\varepsilon}\right)^{\beta}, \qquad A_{\text{ref}} = 0.01, \ \beta \in (0, 1) \tag{13}$$

IoU loss is highly sensitive to center deviations, especially for small objects where even a slight shift can lead to a large IoU drop. However, it lacks explicit penalization for such offset. To address

Table 1: We assess different convolution modules by substituting the first two standard layers in the YOLOv8n (with CIoU loss) and RetinaNet (with Focal loss) detection frameworks. GPConv adopts varying "fanleaf" lengths (e.g., '7, 5' indicates kernel sizes of 7 and 5 for the first and second GPConv layers, respectively). Performance is measured using Precision (P, %), Recall (R, %), and mAP50 (%), while model complexity is represented by the number of parameters (Params, M). The best results are shown in bold, and the second-best are underlined.

| Convolution module | YOLOv8n detection | | | | | | | RetinaNet detction | | | | | |
| | IRSTD-1K | | | SIRST-UAVB | | | Params | IRSTD-1K | | | SIRST-UAVB | | |
| | P | R | mAP50 | P | R | mAP50 | | P | R | mAP50 | P | R | mAP50 |
|---|---|---|---|---|---|---|---|---|---|---|---|---|---|
| Conv | 88.0 | 80.6 | 85.9 | 83.9 | 79.9 | 83.6 | 3.048 | 8.2 | 21.8 | 31.3 | 12.6 | 23.9 | 46.7 |
| DySConv | 87.9 | 79.4 | 85.8 | 87.7 | 83.7 | 88.1 | **3.117** | 23.5 | 35.3 | 66.7 | 36.2 | 48.2 | 84.4 |
| DWConv | 81.2 | 74.4 | 77.6 | 78.5 | 51.1 | 59.6 | **2.660** | 23.4 | 34.5 | 69.2 | 36.8 | 48.0 | 86.4 |
| DSConv | 79.8 | 75.7 | 80.6 | 90.6 | 92.1 | 94.3 | 2.796 | 23.4 | 35.0 | 70.1 | 37.1 | 48.1 | 86.5 |
| WSConv | 86.6 | 83.7 | 86.3 | 88.9 | 89.5 | 92.9 | 3.011 | 24.2 | 35.0 | 69.1 | 18.4 | 30.3 | 50.6 |
| DConv | 90.4 | 80.1 | 79.1 | 88.0 | 84.9 | 89.2 | 2.786 | 23.9 | 35.5 | 69.9 | 28.3 | 38.6 | 76.2 |
| PConv | 87.6 | 82.4 | 86.2 | 91.3 | 89.0 | 91.9 | 2.802 | 21.5 | 35.1 | 64.9 | 40.3 | 48.7 | 87.9 |
| LDConv | 89.5 | 81.2 | 86.1 | 89.6 | 89.2 | 92.7 | 2.791 | 24.1 | 35.2 | 67.9 | 40.4 | 49.2 | 87.9 |
| GPConv(5,5) | 87.0 | **84.6** | 86.8 | 90.4 | 92.2 | 94.4 | 3.048 | 22.6 | 35.0 | 70.1 | **49.4** | 39.3 | **89.1** |
| GPConv(5,7) | 86.9 | 84.1 | 86.5 | 92.0 | **92.4** | **94.9** | 3.048 | **24.9** | **36.0** | **70.6** | 39.7 | **49.9** | 88.6 |
| GPConv(7,7) | **91.8** | 81.7 | **87.6** | **93.1** | 92.2 | 94.7 | 3.048 | 24.7 | 35.6 | 70.2 | 36.3 | 48.3 | 86.5 |

this, we introduce a center distance loss term (Dloss), which weights the normalized center distance $(d/c)$ by the region coverage ratio $(A)$:

$$D_{\text{loss}} = \frac{d^2}{c^2} \cdot \gamma \tag{14}$$

Combining this with the base IoU loss and energy-aware modulation, we define the Infrared Small-Object IoU Loss (IR-SOIoU) as:

$$L_{\text{IR-SOIoU}} = 1 - \text{IoU}^\gamma + \alpha\, D_{\text{loss}} \tag{15}$$

where $\lambda$ is a balancing factor. This formulation explicitly enhances sensitivity to center deviation and object scale, making it well-suited for infrared small-object detection tasks.

## 4 EXPERIMENTS

### 4.1 EXPERIMENTAL SETTINGS

Ablation experiments were conducted on IRST detection models using the PyTorch framework with RTX 5090 GPUs. The models were trained with an input image size of 1280, a batch size of 32, 600 training epochs, an early stopping patience of 70, and a learning rate of 0.01.

### 4.2 COMPARISON WITH OTHER METHODS

#### 4.2.1 COMPARISON OF CONVOLUTION MODULE

Table 1 presents a comparison between GPConv and other convolutional modules. Dynamic snake convolution (DySConv) Qi et al. (2023) and depthwise separable convolution (DWConv) Chollet (2017) focus on enhancing local feature perception, while Distribution shifting convolution( DSC)Nascimento et al. (2019), large selective kernel convolution (WSConv) Zhuang & Lyu (2023), Deformable convnets convolution (DConv) Zhu et al. (2019), Pinwheel-shaped Convolution Yang et al. (2025), and Linear deformable convoluton(LDConv) Zhang et al. (2024) aim to improve robustness to spatial deformations.

In the YOLOv8n detection model, all alternative modules except LDConv have failed to consistently enhance performance. However, LDConv exhibits a low mAP50 and is not able to outperform the GPConv proposed. On the IRSTD-1K dataset, the YOLOv8n model incorporating GPConv (5,7) achieves the best overall performance and the highest average evaluation metric. However, the GPConv (4,3) configuration demonstrates the most balanced improvement while achieving best evaluation metrics. On the SIST-UAVB dataset, GPConv (4,3) delivers the best and most balanced

performance improvement. This demonstrates that a larger GPConv kernel size benefits the detection of larger targets in the IRSTD-1K dataset, while for smaller targets in the SIRST-UAVB dataset, increasing the GPConv kernel size does not yield additional performance improvements. Within the RetinaNet model, GPConv achieves significantly better performance compared to other convolutional modules. The results indicate that a PConv kernel size of 7 in the first layer provides a more effective receptive field, which is crucial for capturing features of small targets. As the feature map resolution and target size decrease during downsampling, a kernel length of 5 in subsequent layers is sufficient, effectively reducing computational overhead while maintaining performance.

The experiments demonstrate that GPConv outperforms other convolutional modules by aligning with the Gaussian distribution of IRST gray levels and effectively expanding the convolutional receptive field. This strengthens the network's capability to extract low-level IRST features with an insignificant increase in parameters.

### 4.2.2 Comparison of Loss Functions

Table 2: Comparison of YOLOv8n using various bounding box losses and the proposed IR-SOIoU loss

| Loss | IRSTD-1K | | | SIRST-UAVB | | |
|------|----------|----------|----------|----------|----------|----------|
| | $P$ | $R$ | mAP50 | $P$ | $R$ | mAP50 |
| CIoU | 88.7 | 83.2 | 87.5 | 93.0 | 89.9 | 93.1 |
| DIoU | 89.7 | 83.4 | 87.5 | 79.6 | 67.9 | 75.0 |
| GIoU | 90.8 | 80.4 | 86.7 | 82.6 | 69.2 | 77.0 |
| IoU | 90.1 | 84.9 | 88.1 | 75.3 | 71.6 | 75.5 |
| WiseIoU | 87.7 | 86.0 | 88.5 | 88.9 | 89.5 | 92.9 |
| SDB | 90.2 | 83.7 | 88.8 | 91.8 | 89.5 | 93.9 |
| IR-SOIoU(0.3) | 90.3 | 86.1 | **89.4** | 93.2 | **93.1** | **95.5** |
| IR-SOIoU(0.5) | 91.3 | 87.1 | 89.3 | **93.9** | 92.1 | 95.1 |
| IR-SOIoU(0.7) | **91.9** | **87.4** | 89.0 | 90.5 | 92.1 | 94.7 |

Tables 2 summarize the performance of various loss functions applied in IRST detection. A comprehensive comparison is conducted among different bounding box-based loss functions, including Complete IoU (CIoU), Distance IoU (DIoU) Zheng et al. (2020), Generalized IoU (GIoU) Rezatofighi et al. (2019), Standard IoU, wiseIoU Tong et al. (2023), SDB ($\delta$) Yang et al. (2025), and the proposed IR-SOIoU loss. Despite its strong performance on the SIRSTUAVB dataset, SDB exhibited a notable performance drop on the IRSTD-1K dataset, indicating limited generalization capability. In contrast, the proposed IR-SOIoU loss demonstrates stable and balanced performance across both datasets, underscoring its robustness for real-world applications characterized by diverse target scales and spatial distributions. Further, the exponential operations in wiseIoU and SDB introduce higher computational costs, while the IR-SOIoU loss remains lightweight and efficient.

From the ablation experiments in Tables 2, the IR-SOIoU loss demonstrates robust and adaptable performance at various threshold settings. While a lower threshold (0.3) yields the highest mAP50 on both datasets—indicating strong overall detection capability—moderate thresholds (0.5) provide a favorable trade-off between precision and recall. In contrast, a higher threshold (0.7) enhances localization precision at the cost of reduced recall, particularly in complex data sets such as SIRST-UAVB. These results highlight the flexibility of IR-SOIoU in accommodating different task requirements.

### 4.3 Ablation Experiments

Table 3 shows that the integration of the proposed GPConv module and the IR-SOIoU loss function consistently improves the performance of various detection frameworks, including YOLOv5n Jocher et al. (2022), YOLOv8n, and YOLOv12 Tian et al. (2025), highlighting the effectiveness and generalization of the proposed approach. Across all the detection models evaluated,ted, the combination of GPCloss and IR-SOIoU Loss consistently achieves the highest mAP50 scores, underscoring its superability to improve detection accuracy. The notable gains in precision and recall, especially within YOLOv12, provide additional evidence of the ability of the proposed method to overcome the inherent limitations of traditional convolutional structures and loss formulations. By improving detection accuracy, stability, and generalization, the GPConv and IR-SOIoU loss functions demonstrate clear advantages and serve as powerful tools for improving detection network

Table 3: Comparative Detection Performance of GPConv and R-SOIoU in Various Models on Two Datasets. CIoU is used as the baseline loss function for detection. In the table, ✓ denotes results obtained using the original method, whereas × indicates those obtained using our proposed approach.

| GPConv | IR-SOIoU | Model | IRSTD-1K | | | SIRST-UAVB | | |
|---|---|---|---|---|---|---|---|---|
| | | | $P$ | $R$ | mAP50 | $P$ | $R$ | mAP50 |
| × | × | YOLOv5 | 86.2 | 82.4 | 85.0 | 78.5 | 62.5 | 71.7 |
| × | ✓ | | 86.7 | **82.5** | 85.0 | 82.3 | **77.5** | 80.5 |
| ✓ | × | | 87.8 | 82.1 | 85.7 | 80.0 | 76.4 | 80.6 |
| ✓ | ✓ | | **87.9** | 82.1 | **86.6** | **86.8** | 76.5 | **81.1** |
| × | × | YOLOv8 | 88.7 | 83.2 | 87.5 | 93.0 | 89.9 | 93.1 |
| × | ✓ | | 90.3 | 86.1 | 89.4 | 93.2 | 92.2 | 94.3 |
| ✓ | × | | 91.8 | 81.7 | 87.6 | 93.0 | 92.4 | 94.9 |
| ✓ | ✓ | | 91.1 | **86.7** | **89.5** | 93.1 | **93.1** | **95.5** |
| × | × | YOLOv12 | 90.4 | 79.1 | 86.6 | 87.7 | 73.2 | 83.9 |
| × | ✓ | | 90.5 | 79.4 | 86.8 | **90.5** | 73.5 | 84.8 |
| ✓ | × | | 91.1 | 79.4 | 84.7 | 84.8 | 79.8 | 84.7 |
| ✓ | ✓ | | **91.9** | **80.2** | **88.7** | 87.9 | **80.5** | **86.1** |

performance. Although the combined approach enhanced YOLOv5 performance relative to baseline, it did not outperform the configuration using GPConv with IR-SOIoU loss alone, indicating that optimal design choices may depend on the characteristics of specific network architectures. In summary, the proposed approach is both robust and highly effective across multiple detection frameworks, underscoring its adaptability and general applicability. Further qualitative analysis of the GPConv and IR-SOIoU loss is presented in Fig.7. GPConv effectively reduces missed detections, while IR-SOIoU loss enhances the detection of weak signals. When combined, they jointly reduce false alarms and improve overall detection robustness.

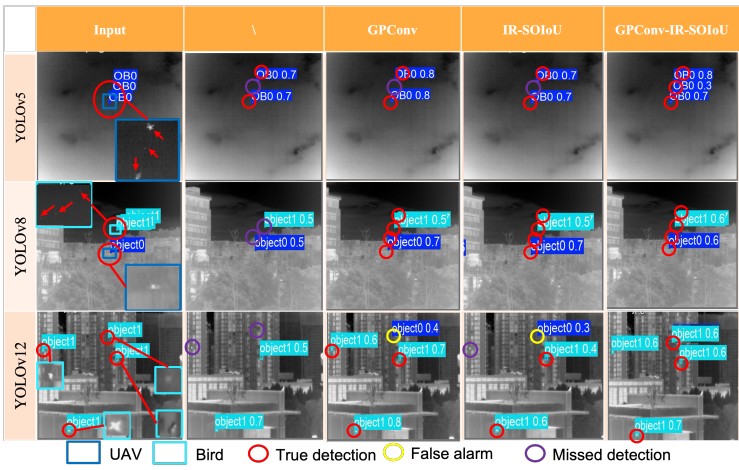

Figure 7: Comparison of Detection Outputs Across Different IRST Models

## 5 CONCLUSION

This paper introduces a plug-and-play GPConv module that integrates Gaussian-prior features, enabling efficient infrared small target detection with an expanded receptive field and minimal parameter overhead. To mitigate the instability caused by IoU fluctuations in label annotations, we further propose the IR-SOIoU loss, a simple yet effective solution that enhances detection stability. Extensive comparisons with existing convolutional modules and loss functions demonstrate that our approach consistently surpasses state-of-the-art methods in both accuracy and robustness. Moreover, the effectiveness and strong generalization capability of the proposed framework have been validated across multiple detection models, underscoring its potential to advance research and applications in infrared small target detection systems.

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

## A  APPENDIX

You may include other additional sections here.

