# OpenReview forum: "Gaussian-Prior Pinwheel Convolution and Region-Energy Loss for Robust Infrared Small Target Detection"
_ICLR.cc/2026/Conference — Submitted to ICLR 2026_

### Official Review · Reviewer_Tyer · 2025-10-22

**Soundness:** 2
**Presentation:** 1
**Contribution:** 2
**Rating:** 2
**Confidence:** 4

**Summary:**

The paper studies infrared small target detection. Inspired by Gaussian characteristics, the authors propose a Gaussian-prior pinwheel convolution and a Infra-Red Small-Object Loss to improve detection performance. Experiments are conducted to show the effectiveness of the proposed methods.

**Strengths:**

The authors try to improve detection performance by modeling the essential characteristics of infrared small targets, which is commendable.

**Weaknesses:**

**Contributions & experiments**
- From the introduction, the core contribution GPCConv is motivated by the assumption that infrared small targets exhibit a gaussian-like shape. However, this motivation is not well substantiated. The paper provides no theoretical basis or empirical evidence to demonstrate that a gaussian-like shape is indeed a general attribute of infrared small targets. Instead, the motivation appears to come from a single observation shown in Fig. 1. A single example is insufficient. Moreover, even in Fig. 1, it is unclear whether the target truly follows a gaussian-like distribution, as the low-response regions likely represent background clutter rather than part of the target itself.
- The proposed Infra-Red Small-Object Loss aims to improve the vanilla IoU loss by introducing an object area ratio factor. However, as shown in eq. 15, it is unclear why assigning higher loss bias to smaller object with works.
- The experiments are insufficient to demonstrate the effectiveness of the proposed Infra-Red Small-Object Loss:
  - For a method focusing on small object detection, the paper lacks discussion of or comparison with common small object detection tasks [1], [2], et al. The authors only evaluate the proposed loss on niche IRSTD tasks, which makes their contributions unconvincing.
  - Table 2 indicates that the proposed IoU loss is sensitive to hyper-parameters. Furthermore, it is not clear whether the compared methods are performed in their optimal settings, as the performance of  DIoU, for example, can also be tuned with hyper-parameters.
I highly recommend that the authors benchmark their method on general small object detection datasets, such as DOTA or the COCO dataset. This would provide a more convincing comparison.

**Presentations**
- The use of notation is unclear:
  - Many symbols are not defined when they are first introduced. Take Eq.1 for example, the authors only give explanation for $W$ which is not even appear in that equation.
  - The same symbol is used for different meanings. For example, in Eq.12, $A$ is used to represent both a ratio and an area. These inconsistencies make the presentation difficult to follow.
- In my opinion, the introduction section is a little long-winded. The authors use two long paragraphs to introduce the details of the IRSTD task. I think a quicker transition to their own discoveries and methods is necessary. It would be better to move the detailed or irrelevant task introduction to the Related Works section or simply discard it.
- *Minor*: The paper incorrectly uses \citet{} in places where parenthetical citations \citep{} are required according to Section 4.1 of the formatting instructions and correct the citation style throughout the manuscript.

**References**

[1] Xu, Chang, et al. "RFLA: Gaussian receptive field based label assignment for tiny object detection." European conference on computer vision. Cham: Springer Nature Switzerland, 2022.

[2] Shi, Zican, et al. "HS-FPN: High frequency and spatial perception FPN for tiny object detection." Proceedings of the AAAI Conference on Artificial Intelligence. Vol. 39. No. 7. 2025.

**Questions:**

- In line 085, what is IoU fluctuation error.
- In line 299, why is the abbreviation for "Infra-Red Small-Object Loss" written as "IR-SOIoU"? I can not find "IoU" in the original term, and the essential word "loss" appears to be missing.

---

### Official Review · Reviewer_egkx · 2025-10-25

**Soundness:** 3
**Presentation:** 2
**Contribution:** 2
**Rating:** 2
**Confidence:** 5

**Summary:**

This paper introduces a novel approach for infrared small target detection (IRSTD) by proposing two key contributions: 1) A Gaussian-Prior Pinwheel Convolution (GPConv) module, designed to replace standard convolutions in the lower layers of a network's backbone to better capture the Gaussian-like spatial distribution of small targets and expand the receptive field with minimal parameter overhead. 2) A Region Energy-Based Loss function that incorporates a dynamic, object-aware weighting factor and a center distance penalty to enhance detection robustness across various target scales, addressing the instability of IoU-based losses for small objects. The authors integrate these components into the YOLOv8n and RetinaNet frameworks and demonstrate significant and consistent performance improvements on the public IRSTD-1K and SIRST-UAVB datasets.

**Strengths:**

- The proposed components are presented as plug-and-play modules, making them easy to adopt and integrate into other existing detection frameworks. This practicality increases the potential impact of the work.
- The paper demonstrates consistent and significant performance gains by integrating the proposed components into two distinct object detection frameworks (YOLOv8n and RetinaNet). The evaluation on two public benchmark datasets (IRSTD-1K and SIRST-UAVB) validates the effectiveness and generalizability of the approach.
- The paper does a good job of explaining the rationale and mechanics behind the GPConv module and the Region Energy-Based Loss function. The explanations are buttressed by diagrams (albeit somewhat vague) and mathematical equations.

**Weaknesses:**

- In the paper, at Line 300, it is claimed that "As depicted in Fig. 6, the Infra-Red Small-Object Loss (IR-SOIoU) demonstrates fluctuations reaching up to 86%." Nevertheless, such fluctuations of up to 86% are not discernible from the figure. To clarify this discrepancy, the author should provide a detailed explanation of Fig. 6.
- The paper presents a novel GPConv structure, yet it is short of experimental results comparing this structure with the transformer.
- It should be noted that the “neuron-level 3D attention mechanism” is repeatedly put forward as a key contribution , but unfortunately, it does not appear at all in the Methodology section. To enhance the comprehensiveness and clarity of the paper, the author is advised to clearly elaborate in the related section what this mechanism entails, how it operates, and how it is specifically implemented.
- Nearly all the citation formats of the references in the paper are incorrect. It is crucial to be aware of the difference between `\citep` and `\citet`. In the ICLR LaTeX template, the \cite command specifically denotes the inline citation form, which is distinct from that of other conferences.
- The arrangement of the pictures appears to be somewhat ill-conceived, resulting in the waste of a substantial amount of space. Specifically, this issue is evident in Figure 1, Figure 2, Figure 3, and Figure 5.
- The backbone networks, namely YOLOv8n and RetinaNet, are somewhat outdated. It would be advisable to conduct additional experiments on more advanced networks.

**Questions:**

1.  Would you kindly provide detailed explanations for Figure 6? It seems that this figure visualizes the IoU fluctuation.
2.  The abstract and introduction list a "neuron-level 3D attention mechanism" as one of the paper's contributions. Could you please provide the missing methodological details for this mechanism?

---

### Official Review · Reviewer_4QYU · 2025-10-25

**Soundness:** 2
**Presentation:** 1
**Contribution:** 2
**Rating:** 2
**Confidence:** 4

**Summary:**

This paper addresses the challenges of infrared small target detection, such as tiny object size and low signal-to-noise ratio, by proposing GPConv and IR-SOIoU. GPConv enhances feature extraction and receptive field expansion through a Gaussian prior and pinwheel-shaped design, while IR-SOIoU improves localization stability via dynamic weighting and a center distance penalty. Experiments on IRSTD-1K and SIRST-UAVB datasets demonstrate the method’s effectiveness and generalization across multiple detection frameworks.

**Strengths:**

# **Strengths**

1. **Originality:** GPConv explicitly embeds the physical prior of infrared small targets (Gaussian distribution) into the convolution structure, representing a meaningful and customized enhancement of standard convolution for a specific modality. The IR-SOIoU loss function provides a clear modeling approach to address the IoU instability issue in small target detection, and the dynamic weighting design is well-motivated and reasonable.

2. **Quality:** The experimental design is rigorous, covering multiple backbone networks (YOLOv5/v8/v12, RetinaNet), two public datasets, and comprehensive comparisons of various convolution modules and loss functions. The ablation studies (Table 3) effectively verify the contribution of each component, and the parameter analysis (Eq. 10) reflects consideration of computational efficiency.

3. **Clarity:** The paper is clearly structured, with rich figures and tables, and provides detailed explanations of both the methodology and experimental analysis.

4. **Significance:** Infrared small target detection is of great importance in both military and civilian applications. The proposed GPConv and IR-SOIoU modules are “plug-and-play” in nature and can be easily integrated into other detection frameworks, contributing positively to the practical advancement of IRSTD.

**Weaknesses:**

# **Weaknesses**

1. **Limited Innovation Scope:** The “pinwheel-shaped structure” and Gaussian-weighted design in GPConv partially overlap with prior works such as *PinwheelConv (AAAI 2025)* and Gaussian kernel-based weighting strategies. The paper should more clearly delineate the distinctions and improvements compared to these earlier methods.

2. **Insufficient Methodological Details:** The 3D attention mechanism is mentioned only in the abstract but lacks detailed architectural diagrams, formulas, or ablation studies in the main text, making it difficult to assess its contribution.

3. **Shallow Experimental Depth:**
   - All experiments focus on single-frame detection tasks, without evaluation on sequential scenarios (e.g., video-based or multi-frame fusion).
   - The paper does not provide an analysis of the trade-off between inference speed (FPS), model complexity, and performance, limiting the assessment of practical usability.

4. **Limited Explanation of the Loss Function:**
   - The rationale behind the selection of parameters γ(A) and β in IR-SOIoU is not discussed, and there is no analysis of how different β values affect performance on small vs. large targets.
   - The paper lacks a gradient-level comparison with other recent loss functions such as **WiseIoU** or **SDB Loss**; visualizing gradient curves in the appendix would strengthen the analysis.

5. **Insufficient Comparison with Specialized Networks:** Although comparisons are made with various convolution modules, the paper omits comparisons with recent specialized small-target detection networks (e.g., **DNANet**, **ISTDet**), which would help contextualize performance.

6. **Empirical Loss Design:** The hyperparameters (e.g., β, α) in IR-SOIoU appear empirically chosen, with no theoretical justification or sensitivity analysis provided. Including such analysis would enhance the reliability of the design.

7. **Unassessed Computational Efficiency:** Although the paper claims a “marginal parameter overhead,” it does not report inference speed (FPS), parameter count, or FLOPs comparison, making it difficult to evaluate real-time performance.

**Questions:**

# **Questions**

1. **On the Consistency of Claimed Contributions:**
The title and abstract emphasize three main contributions (Gaussian-Prior Pinwheel Convolution, Region Energy-based Loss, and 3D Attention). However, the introduction only elaborates on the first two, without clearly describing the role and innovation of the “3D Attention” module. Could the authors clarify its design details and how it integrates into the overall framework?

2. **On the Relationship Between GPConv and Prior Work:**
The “pinwheel-shaped structure” and Gaussian-weighted concept of GPConv appear similar to prior works such as *PinwheelConv (AAAI 2025)*. Please clearly articulate the key differences and improvements in terms of structural design, computational process, or theoretical motivation.

3. **On Formula and Figure Consistency:**
Some inconsistencies exist between symbols used in formulas and figures (e.g., K in Eq. (1) versus W in the explanation). Additionally, in Figure 4, the feature map dimensions and the mask shapes do not align. Please review and unify the notation and visual annotations to avoid confusion.

4. **On the Interpretation of Experimental Results:**
The source and configuration of GPConv(4,3) in Table 1 are unclear, and some highlighted values appear incorrect. Please verify the correctness of these results and clarify the rationale behind the comparisons and interpretations.

5. **On Comparisons with Specialized Networks:**
The paper does not include comparisons with methods specifically designed for infrared small target detection. The current experiments mainly verify the advantage of GPConv over other convolution modules within YOLO and RetinaNet architectures. Including such specialized baselines would strengthen the empirical validation.

---

### Official Review · Reviewer_tGiT · 2025-10-27

**Soundness:** 2
**Presentation:** 1
**Contribution:** 1
**Rating:** 2
**Confidence:** 3

**Summary:**

This paper builds upon the observation that the grayscale values of infrared small targets approximately follow a Gaussian distribution. Based on this observation, the authors propose a Gaussian-prior-based attention-enhanced pinwheel convolution. They also design a Region Energy-Based Dynamic Loss to improve infrared small target detection performance.

**Strengths:**

- The idea of introducing a Gaussian prior to model the intensity distribution of small infrared targets is intuitively reasonable and potentially beneficial.
- The paper targets an important and challenging task.

**Weaknesses:**

- The proposed method appears highly similar to the paper "Pinwheel-shaped Convolution and Scale-based Dynamic Loss for Infrared Small Target Detection" published in AAAI 2025. The new contribution seems to be a relatively simple extension of that work, and there are also notable textual similarities between the two manuscripts.
- The writing quality is poor, making the paper difficult to read. Many sections lack clarity, and several definitions and notations are ambiguous or missing.

**Questions:**

- Line 74: The statement "can be further subdivided into detection-based frameworks"  Shouldn't segmentation-based frameworks also be included?
- Line 84: The authors claim that "both bounding box (BBox) and mask annotations suffer from considerable IoU fluctuation errors" but Figure 2 only shows bounding boxes. In addition, Figure 2 is difficult to interpret and fails to effectively illustrate the IoU fluctuation errors.
- Line 88: The acronym IR-SOIoU appears for the first time here. Its full name should be provided rather than using the abbreviation directly.
- Contribution 2: The statement "We introduce a region energy-based dynamic loss…" refers to a concept that appears only in the abstract but not in the introduction. The authors should ensure consistency in presentation across sections.
- Contribution 3: The sentence "We integrate GPConv and IR-SOIoU Loss into both bounding box formats…" is unclear. What exactly are "both bounding box formats"? Please clarify.
- Related Work: The authors discuss existing datasets, yet the paper itself does not provide any dataset-related contribution.
- Line 191: The sentence starting with "In the first layer of" seems incomplete. Is there missing content?
- Equation (1): The variable K is introduced without explanation.
- Figure 4: The term $\hat{W}$ is mentioned in the text, but it does not appear in the figure. Only $W'$ and $W''$ are shown. How is $\hat{W}$ actually used?
- Equation (14): The meanings of $d$ and $c$ are not defined, and it is unclear where the region-energy term $W_e$ is applied.
- The image sizes in the paper are not well adjusted. The text within several figures is too small to read clearly, which significantly affects readability.
- It is recommended that the authors also incorporate their proposed approach into a segmentation-based framework to further evaluate its effectiveness and generalization.

---

### Meta-Review · Area_Chair_nyNa · 2025-12-20

**Summary:**

This paper proposes a novel approach for Infrared Small Target Detection (IRSTD) by introducing three main components: Gaussian-Prior Pinwheel Convolution (GPConv), a Region Energy-Based Loss (IR-SOIoU), and a Neuron-Level 3D Attention mechanism. The authors claim these components, when integrated into modern detection frameworks, achieve state-of-the-art results.

The paper received four reviews, resulting in a unanimous and strong negative consensus with scores of **[2, 2, 2, 2]**. Crucially, the authors **did not submit a rebuttal** or engage in any discussion with the reviewers. Consequently, all identified issues remain unaddressed.

The paper is a clear reject due to a combination of fundamental flaws, including a lack of substantiated motivation for its core contribution, significant overlap with prior work raising novelty concerns, missing methodological details for a claimed key contribution, and severe presentation issues. The unanimous decision from the reviewers, combined with the authors' lack of engagement, solidifies this recommendation.

**Reviewer Concerns:**

Since no author rebuttal was provided, all concerns are considered outstanding. The decision to reject is based on the cumulative weight of these unaddressed issues.

**Reviewer Scores:**

Since no rebuttal was submitted, no reviewer had the opportunity to participate in a discussion phase.

---

### Decision · Program_Chairs · 2026-01-26

Reject